# Implementation of Image-Guided Brachytherapy for Pediatric Vaginal Cancers: Feasibility and Early Clinical Results

**DOI:** 10.3390/cancers14133247

**Published:** 2022-07-01

**Authors:** Mario Terlizzi, Véronique Minard, Christine Haie-Meder, Sophie Espenel, Hélène Martelli, Florent Guérin, Cyrus Chargari

**Affiliations:** 1Department of Radiation Oncology, Gustave Roussy Cancer Campus, 94800 Villejuif, France; mario.terlizzi@gustaveroussy.fr (M.T.); chaiemeder@gmail.com (C.H.-M.); sophie.espenel@gustaveroussy.fr (S.E.); 2Department of Pediatric and Adolescent Oncology, Gustave Roussy, Université Paris-Saclay, 94805 Villejuif, France; veronique.minard@gustaveroussy.fr; 3Department of Pediatric Surgery, Paris-Saclay University, Assistance Publique-Hôpitaux de Paris, Bicêtre Hospital, 94270 Le Kremlin Bicêtre, France; helene.martelli-ext@aphp.fr (H.M.); florent.guerin@aphp.fr (F.G.)

**Keywords:** gynecologic cancer, primary vaginal tumor, rhabdomyosarcoma, malignant germ cell tumor, pediatric, brachytherapy

## Abstract

**Simple Summary:**

Brachytherapy is one of the cornerstones of the treatment of pediatric vaginal tumors in combination with surgery and chemotherapy from a conservative perspective. In this retrospective series, we present our experience with 3D pulsed dose rate brachytherapy for the treatment of children with vaginal tumors. Our results show that this treatment has good compliance and provides an excellent local control rate. The toxicity rate is also favorable, with gynecological toxicities being the most frequent. Given the rarity of these diseases, their management should be entrusted to expert centers.

**Abstract:**

*Background:* Brachytherapy (BT) has a major role in pediatric cancers of the lower genital tract, as part of a multimodal organ conservative strategy. Scarce data are available on the location of image-guided BT. *Methods:* Medical records of all consecutive girls treated in our center between 2005 and 2020 for a vaginal tumor with exclusive image-guided PDR-BT were retrospectively examined, with a focus on treatment parameters, patient compliance, and clinical outcome, including analysis of local control, survival and late toxicity rates. *Results:* Twenty-six patients were identified, with a median age of 25 months. Histological types were rhabdomyosarcoma, malignant germ cell tumor (MGCT) and clear cell adenocarcinoma in 18 (69%), 7 (27%) and 1 (4%) patients, respectively. Ten (33%) patients had prior surgery and 25 (96%) received chemotherapy prior to BT. The median prescribed dose was 60 Gy through pulses of 0.42 Gy. Global compliance was satisfactory, but three (12%) patients required replanning because of applicator displacement. After a median follow-up of 47.5 months, one patient with MGCT referred for salvage treatment of a local recurrence had a local and metastatic relapse. The local control rate probability was 96% at the last follow-up. Late toxicity rates ≥ grade 2 and ≥ grade 3 were reported in 23% and 11%, respectively, with gynecological toxicities being the most frequent side effect. Two patients required dilatation for vaginal stenosis. *Conclusions:* PDR-BT allowed similar local control compared to the historical low-dose rate technique. An indirect comparison suggests fewer treatment-related toxicities by integrating image guidance and optimization capabilities, but longer follow-up is necessary. Due to the rarity of the disease and the technical aspects of BT in these very young patients, referral to specialized high-volume centers is recommended.

## 1. Introduction

Gynecologic tumors are rare in childhood, estimated at less than 5% of all pediatric cancers. Main histologies are represented by rhabdomyosarcoma (RMS), malignant germ cell tumor (MGCT) and clear cell adenocarcinoma. In vaginal tumors, a dual challenge is to achieve a cure while preserving fertility and avoiding mutilating treatment. In this context, local treatment modalities are highly challenging.

While surgery was the cornerstone of vaginal RMS treatment until the 1980′s, several studies showed that a multimodal strategy combining neo-adjuvant chemotherapy, radiotherapy (RT) with or without brachytherapy (BT), and conservative surgery allowed high long-term overall survival rates, greater than 90% [1]. For vaginal botryoid RMS, local treatment is necessary only in the case of residual tumors after chemotherapy [2]. Vaginal RMS has a good prognosis, and it is essential that the delivered treatments have the fewest long-term sequelae. Conservative surgery through partial colpectomy is frequently not feasible, especially for multifocal tumors or when the lower third of the vagina is involved. External Beam Radiation Therapy (EBRT) may lead to major sequelae in these very young patients (median age: 2 years), including bone growth disorders, bladder dysfunction, gastrointestinal toxicities, and uterine dysfunction. BT is an ideal irradiation technique to avoid the long-term side effects of external irradiation. It is also an alternative to surgery in selected patients with MGCT or clear cell carcinoma who cannot be treated with conservative surgery, especially if there is para-vaginal involvement.

Given the rarity of pediatric vaginal tumors, few studies have reported the oncological outcomes of patients treated with BT, and the few available data have reported on the low-dose-rate (LDR) technique, with 2D treatment planning and marginal optimization capabilities in terms of dose distribution. Institutional series have shown excellent local control, but genital sequelae are significant. In the past decade, developments in stepping source technology and 3D-guide treatment planning have led to significant improvements in local control and functional results among adult patients with cervical or vaginal cancers treated with BT. However, scarce data are available on the application of 3D-guided treatments in the pediatric population. In this series, we examined the outcome of girls treated with image-guided pulsed dose-rate BT (PDR-BT) for a vaginal tumor with curative intent.

## 2. Materials and Methods

### 2.1. Patients

All consecutive children who received 3D PDR-BT as part of their treatment with curative intent for a vaginal tumor at Gustave Roussy, Villejuif, France, were retrospectively analyzed. Patients treated with external radiotherapy in addition to BT were excluded from the analysis. The children were treated following a multidisciplinary tumor board (MTB) discussion and after obtaining their parent informed consent. The indication for vaginal brachytherapy relied on a careful discussion weighting other possible treatments, including surgery, to achieve the highest probability of cure without mutilation and without jeopardizing the possibility of further fertility. Tumors were classified according to their location: upper, middle and lower vagina. Data regarding patients (age), tumor (initial characteristics) and treatments (chemotherapy, surgery and BT characteristics) were collected. This study is part of a broader project examining the outcome of patients treated with pediatric BT in our center and obtaining ethics committee approval (reference 2021-28).

### 2.2. Treatment

Patients received BT as part of an organ-conservative strategy. Patients with RMS or MGCT were treated with upfront chemotherapy. RMS tumors were treated following the EpSSG (European Soft Tissue Sarcoma Study Group) RMS 2005 protocol and MGCT were treated with a platinum-containing chemotherapy regimen. Patients had a gynecological examination under general anesthesia at diagnosis or after 1–3 cycles if tumor regression was necessary to allow a gynecological examination associated with a vaginal impression to show tumor extensions at diagnosis [3].

The indication for performing local treatment depended on tumor histology. For RMS, only patients with residual tumors after six cycles received a local treatment, which was scheduled after a minimum of 6 cycles of chemotherapy. When the residual tumor involved the cervix, surgery was prioritized as the first intent treatment (upper colpectomy +/− trachelectomy) to avoid irradiation of the uterus. BT was, however, indicated when organ-sparing surgery was not feasible (e.g., tumor involving the paravagina and/or the parametrium and/or multifocal and/or involving the two lower thirds of the vagina). After BT completion, chemotherapy was continued as per the protocol for up to nine cycles. Patients with vaginal RMS having complete clinical and radiological responses confirmed by biopsies after first-line chemotherapy after six cycles did not receive any local treatment and were considered for close clinical and radiological surveillance. For MGCT, local treatment was systematically performed after chemotherapy was completed. Conservative surgery (tumorectomy) was considered the mainstay of treatment in these patients and BT was proposed in situations where upfront surgery would have led to significant functional impairment or was not technically possible or for treatment of local relapses. For clear cell carcinoma, the treatment choice between surgery and BT depends on tumor topography. Surgery was prioritized for tumors involving the upper third of the vagina and the cervix, and BT was proposed only if a conservative surgical approach with safe margins was not feasible or for patients having local relapse following surgery and without previous irradiation.

### 2.3. Brachytherapy Delivery

A few days prior to BT implantation, magnetic resonance imaging (MRI) with a T2 axial sequence was performed to assess tumor response and plan the BT implant. In addition, the patients underwent a clinical examination under general anesthesia, and another vaginal impression was performed to show residual tumor extents. From that vaginal impression, a personalized vaginal mold applicator was built. The methodology applied at our institution for pediatric tumors is derived from that for adults, where the material used is adapted to the anatomy of both the children and the tumors (shape and vaginal extents). For the intracavitary technique, the numbering and positioning of the catheters were decided on an individual basis. A washing catheter was also fixed to the vaginal mold applicator to allow vaginal irrigation during treatment. Technical aspects of vaginal mold applicators have been previously described [3]. At the time of this procedure, a temporary ovarian transposition was performed to minimize the radiation exposure of the ovaries [4].

A few days after the gynecological examination, the vaginal mold applicator was inserted under general anesthesia together with a Foley urinary catheter. Gold fiducial markers were implanted into the vaginal wall to mark the extent of the tumor. For patients with significant residual disease, a conservative debulking tumorectomy could be done without resection of the vaginal wall to allow vaginal mold insertion. Interstitial catheters were inserted following a free-hands technique if judged necessary to achieve good coverage of the treated volume, especially if the tumor thickness was more than 5 mm. In cases of associated cervical extent or for tumors involving the fornices, an intracervical catheter was inserted and sutured to the vaginal mold applicator. The vaginal mold applicator was sutured to the vaginal wall to avoid displacement during treatment.

After implantation, imaging scans for treatment planning were performed without anesthesia. Computed tomography (CT) with scan acquisition was performed in the supine position with an applicator in situ, with a slice thickness of 1.5 mm. The target volume was delineated on the CT scan, taking into account clinical findings and data from pre-BT MRI or from an MRI with an applicator in place, axial T2 sequence fused with a CT scan. A CT scan was always necessary for treatment planning, as these very thin catheters placed within the vaginal mold are not well visualized on MRI. Catheters were digitized, and computerized dosimetry was performed. A total physical dose of 60 Gy was prescribed to 90% of the clinical target volume (CTV). The CTV was defined as the residual tumor, taking into account data from clinical examination, position of the gold fiducials and MRI findings. Areas with residual hyperintense signal intensity on T2-weighted MRI were included, as well as areas with abnormal thickening or irregularity within the initial tumor extension (before chemotherapy) [2]. For postoperative cases, the vaginal wall (usually around 2–3 mm thickness) was delineated in regard to the area with positive margins, taking into account surgical reports and pre-operative MRI. The following organs at risk were delineated for dose reporting and optimization: rectum, bladder, sigmoid colon, transposed ovaries, and uterine corpus. The noninvolved part of the vagina was retrospectively delineated for study purpose and was defined as the whole vagina minus the CTV.

Dose was initially prescribed to the external part of the vaginal wall at the mid-level of the implant. Then, dose optimization was performed through dwell times and position modifications to keep organs at risk doses as low as possible, without specific dose/volume constraint. It was aimed that the 100% isodose did not overlap with organs at risk volumes, especially at the level of the anterior rectal wall and that volumes of the vaginal wall receiving 150% isodose were as minimal as possible. An example of dose distribution is shown in Figure 1. Irradiation was delivered with an Iridium-192 source, and the total dose was delivered through hourly pulses of 0.42 Gy per pulse to keep the daily dose at 10 Gy. Doses to the tumor and to organs at risk were reported after conversion into EQD2 (equivalent dose per 2 Gy fractions, α/β value = 10 Gy for CTV and 3 Gy for organs at risk). Usual precautions for pediatric PDR treatments were followed, as previously reported for bladder prostate RMS treatments [5]. Briefly, the parents had information on radioprotection rules. They were not allowed to stay in the brachytherapy department overnight. A daily examination was done by dedicated radiotherapy technicians to ensure that there was no applicator displacement. In case of any doubt, radiographs or CT scans were acquired to check for the applicator position. A pediatrician visited the patient daily and more frequently, if specifically required, to prescribe analgesics and soft-conscious sedation if required.

### 2.4. Statistics and Follow-Up

After treatment completion, the patients were followed according to pediatric protocols. A complete clinical examination with vaginoscopy was performed every 4 to 6 months during the first 2 years and every 6 to 8 months thereafter up to 5 years. MRIs were performed following the same schedule and were reviewed for MTB. Late toxicities, defined as any toxicity occurring more than three months following BT, were recorded and classified according to the Common Terminology Criteria for Adverse Events (CTCAE v5.0) [6]. The first sites of relapse were examined and classified into local (vaginal), regional (inguinal or pelvic lymph nodes) or distant (metastases). Data inclusion was performed from January 2005 to December 2020 with a cut-off analysis in January 2022. Statistical analyses were carried out with *R software, version 3.6.3, RStudio Inc., Boston, USA*.

## 3. Results

### 3.1. Patients

The selection process is presented in Figure 2. Between 2005 and 2020, a total of 30 children were treated in our institution for vaginal tumors. Three patients with locally advanced clear cell carcinoma and one patient with synchronous bladder alveolar RMS with lymph node extension had also received external radiotherapy and were therefore excluded from the analysis.

Among the 26 remaining children, the median age was 25 months (range 7 months–12 years). Twelve (46%) patients were referred from abroad and 14 (54%) were referred from national centers. Eighteen were referred for primary treatment and eight were referred for treatment for vaginal relapse. Only one patient, with a MGCT, presented with lymph node involvement at diagnosis and none had distant lesions (M0). Tumor localization was upper, middle and lower vagina for 15 (58%), three (12%) and eight (31%) patients, respectively. Histology was RMS in 18 (69%) patients, MGCT in seven (27%) patients, and clear cell adenocarcinoma in one (4%) patient. Among the patients with RMS, 15 (83%) and three (17%) had botryoid and embryonal histological subtypes, respectively. Seventeen tumors (65%) measured 5 cm or more and nine (35%) measured less than 5 cm.

Ten patients (33%) had prior surgery. Five of them had previously undergone partial colpectomy (three were referred for adjuvant treatment because of microscopically involved margins and two were referred because of a local relapse). Four patients had previous tumorectomy (three required debulking tumorectomy to allow the placement of the vaginal mold applicator, and one was referred for treatment of a relapsed MGCT after initial tumorectomy). One patient had a partial vulvectomy for an RMS and was referred for treatment of a para-urethral relapse. A total of 25 (96%) patients received chemotherapy (median cycles 8, range 4–15) prior to the BT procedure. Only the patient with clear cell carcinoma received brachytherapy without previous chemotherapy. A surgical ovarian transposition was performed on 25 (96%) patients. One patient with a tumor involving the very distal part of the vagina did not require ovarian transposition after pre-planning, showing that the ovarian dose was very low (<1 Gy). The patient characteristics are detailed in Table 1.

### 3.2. Treatment Characteristics

BT technical characteristics (implantation, doses and volume) are presented in Table 2. All patients had an intracavitary application. Vaginal application was performed with a molded applicator, including a median number of three catheters (range: 2–4). Two (8%) patients were treated with a combined intracavitary/interstitial (free hands implantation) technique (up to three interstitial catheters were implanted). Two patients required the placement of an intra-uterine catheter to treat the cervix. Dosimetry was performed on CT scan imaging for 14 patients, taking into account pre-brachytherapy MRI to guide target volume delineation, and it was performed on MRI with applicator in situ in 12 patients. Parts of the cervix were included in the CTV in eight (31%) patients who presented with an upper vaginal tumor extending to the external portion of the cervix. A median physical dose of 60 Gy (range 50–60.06 Gy) with a median number of pulses of 143 (range 100–150) was prescribed.

The median CTV volume was 2.55 cm^3^ (range 0.36–11.98 cm^3^). The median of the minimum doses delivered to 90% and 80% of the CTV (D90 and D80, respectively) in EQD2 were 51 Gy_EQD2_ (9.3–71.4 Gy_EQD2_) and 60.9 Gy_EQD2_ (0.7–79.3 Gy_EQD2_), respectively. The median of the minimal doses delivered to the most irradiated D1cc parts of the bladder, rectum and sigmoid were 27.3 Gy_EQD2_ (0.4–61.4 Gy_EQD2_), 35.0 Gy_EQD2_ (0.3–67.7 Gy_EQD2_) and 4 Gy_EQD2_ (0.2–28.3 Gy_EQD2_), respectively. The median dose to the uterine corpus was 9.8 Gy_EQD2_ (0.2–98.1 Gy_EQD2_). The median Total Reference Air Kerma was 0.91 cGyh^−1^ m^−1^.

### 3.3. Local Control and Survival

With a median follow-up of 47.5 months (range 1 month–15 years), one patient presented with a local and metastatic (lung) relapse. She was previously treated with salvage BT for local recurrence of a MGCT treated one year before with surgery and chemotherapy. No patient with RMS or clear cell carcinoma relapsed. All patients were alive at the time of analysis. Due to the 96% disease control rate at the last follow-up, no statistical analysis was conducted to identify the causes for failure.

### 3.4. Treatment Compliance

Three patients (12%) expulsed the vaginal mold during the treatment and required re-implantation and dosimetry. For two of them, the material showed displacement during movement and for the other, the material was expulsed during a defecation effort. It was necessary to reposition the vaginal mold applicator during hospitalization due to difficulties with bladder emptying in two (8%) patients (compression of the urethral catheter by the vaginal mold). No other acute complications were observed, and all patients could complete their brachytherapy at the prescribed dose.

### 3.5. Late Toxicities

A total of six patients (23%) presented with grade ≥ 2 late toxicity, including three patients (11%) having grade 3 late complications (Table 3). Gynecological toxicity was the most common consequence of treatment. Two patients had vaginal grade 3 stenosis (stenosis requiring at least one dilatation) diagnosed with a median time interval of 23 and 31 months from BT, respectively, including one patient also having grade 3 proctitis requiring coagulation with plasma argon. Two patients more than 4 years at last follow-up had late grade 2 urinary incontinence.

## 4. Discussion

Several retrospective series, including those from our group, showed the place of BT for the treatment of genito-urinary malignancies. BT offers the possibility of delivering a highly conformational and localized dose, reducing the dose received by the surrounding healthy tissues and thus decreasing the risk of long-term functional impairment [3,7,8,9,10]. For vaginal tumors, BT is therefore classically preferred over EBRT. A large international series from four cooperative groups (Children’s Oncology Group, COG; International Society of Pediatric Oncology-malignant Mesenchymal Tumor, SIOP MMT; Italian Cooperative Soft Tissue Sarcoma Group, ICG and European Pediatric Soft Tissue Sarcoma Group, EpSSG), including 237 patients treated for localized vaginal/uterine RMS was recently published. Among 160 (68%) patients with vaginal tumors of whom 25 (37%) received BT as a part of their treatment, 10-year event-free and overall survival were 74 and 94%, respectively [11].

Reducing the long-term sequelae of treatments is one of the main challenges in pediatric oncology. Especially for gynecological tumors, several studies have reported that treatments may have a major impact on physical (digestive and urinary), sexual (fertility) and psychological functions [12,13]. Over the past decades, there has been progressive refinements in BT indications and techniques in order to maintain high local control probability while decreasing the risk of severe complications. At the time of LDR techniques, a major change in treatment strategy emerged in the 90′s after publication that it was possible to use BT to irradiate only the tumor residuum and not initial tumor extension, without jeopardizing disease control probability. Such changes led to significantly fewer complications among children treated for vaginal RMS [2,7,11]. This significant change in BT target volumes showed comparable survival outcomes, while limiting late toxicities [11]. Thus, it was shown that vaginal/urethral sclerosis and stenosis rate was 75% for patients treated before 1990 versus 20% after 1990, in relationship with the decrease in treated volume from pre-chemotherapy volume to post-chemotherapy residual volume [14]. In another retrospective study of 42 females with RMS genital tract (81% vagina) treated at Gustave Roussy with BT, Levy et al. reported a late-effect rate in 76% of patients (72% G1-2 and 28% G3-4) after a 15-year follow-up. Prognostic factors for the total number of all grade and G3-4 late effects were the BT period (before 1990), the cumulative dose (>60 Gy), the maximal dose delivered to ovaries (>1.7 Gy) and the BT volume (>11 cm^3^) [9].

More recently, the integration of image-guidance and stepping source technology has provided the possibility of better adapting treatment delivery and dose distribution on an individual case-per-case basis. It is now possible to accurately document doses to organs at risk and minimize those through optimization of dwell and time positions within the implanted volume. PDR-BT indeed shares some of the radiobiological advantages of LDR (in terms of normal tissue sparing), but it also gives the possibility to perform dose optimization. In addition, the integration of 3D imaging based with MRI-based BT helps to better identify the residual disease that should be treated compared to historical 2D radiographs-based treatments. However, only scarce data are available on the results of such 3D treatments in children. The specificity of this study is the systematic use of a modern BT technique based on 3D imaging and dose optimization to achieve very high local control probability and minimize the risk of severe late toxicities. This is the first report of dosimetric and clinical data for patients treated with image-guided PDR-BT for vaginal tumors. After a median follow-up of 47.5 months, the control rate achieved was high (96%), with only one patient having both local and distant recurrences. Moreover, no patients with vaginal RMS experienced disease relapse. These results confirm the excellent control rate and survival of vaginal pediatric tumors when treated with a conservative strategy. This is consistent with the results of series reporting on LDR technique use (Table 4).

To our knowledge, this is the first series to describe a methodology for image-guided BT in pediatric vaginal tumors with a definition of a CTV, as well as to report dose/volume parameters in these patients. In our study, the mean treated CTV volume was 3.75 cm^3^ and median D_90_CTV was 51 Gy_EQD2_, which means that a physical dose prescription of 60 Gy was necessary to achieve a curative dose of at least 50–50.4 Gy to 90% of residual disease. We observed that 23% of patients presented with late ≥ grade 2 toxicities, of which only two had ≥ grade 3 gynecological toxicities (vaginal stenosis requiring dilatation). This relatively low rate of severe complications compared favorably with data from a previous series from our center at the time of 2D treatments. The ability to perform dose optimization probably contributed to the relatively low incidence of severe events compared to historical data. We also confirmed the feasibility of PDR-BT in these very young patients with acceptable compliance. Therefore, it is possible to exploit the theoretical radiobiological advantages of this technique over high-dose BT. However, we observed that it was necessary to re-implant or change the position of the BT material in five patients. This highlights that pediatric BT should be only performed in high-volume centers with trained medical and paramedical teams familiar with adult BT procedures but also able to manage these unpredictable events [15].

Our study has several limitations. Though median follow-up is enough to confirm excellent local control rates with image-guided BT, follow-up still remains too short to draw definitive conclusions on either quality of life or pubertal development. A longer follow-up with robust quality-of-life data would be important in order to analyze more accurately the development of long-term sequelae. In addition, the total number of patients is low due to the scarcity of the disease, despite our Institute is a referral center for such diseases. A preplanned prospective follow-up with complete questionnaires exploring several items (LENT-SOMA or SF-36v2) remains necessary for further recommendations. In addition, follow-up remained quite short, in line with the fact that 16/26 (62%) patients were treated in 2015–2020, with three of them requiring vaginal dilatations. Additional follow-up is still required to assess the impact on uterine growth and the potential need for reconstructive surgery. Finally, we chose to exclude patients who received chemoradiation because of locally advanced diseases associated with a worse prognosis and because EBRT treatment is associated with different toxicities that would have biased the interpretation of the results.

## 5. Conclusions

In conclusion, our series, mainly including vaginal RMS, shows that a multimodal organ conservative strategy, including PDR-BT, can be used to achieve excellent oncological outcomes for pediatric vaginal cancers. Because of the scarcity of the disease and the long-term impact of treatments, it appears essential to refer patients to high-volume specialized centers with expertise in adult image-guided adaptive BT.

## Figures and Tables

**Figure 1 cancers-14-03247-f001:**
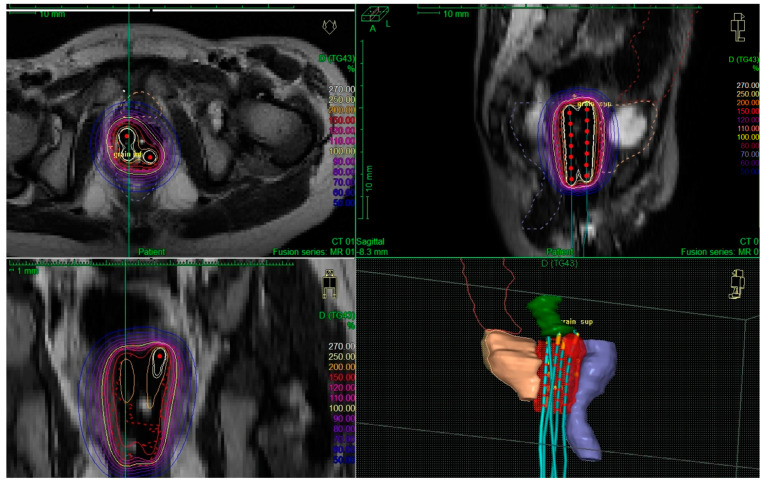
Example of an image-guided treatment for a 2-year-old girl with vaginal rhabdomyosarcoma. At the time of brachytherapy, there was multifocal residual disease involving the left and posterior parts of the vagina. An intracavitary procedure was performed with a vaginal mold applicator (four catheters). The CTV is red, the bladder is orange, and the rectum is blue. The 100% isodose (60 Gy isodose) is shown in yellow.

**Figure 2 cancers-14-03247-f002:**
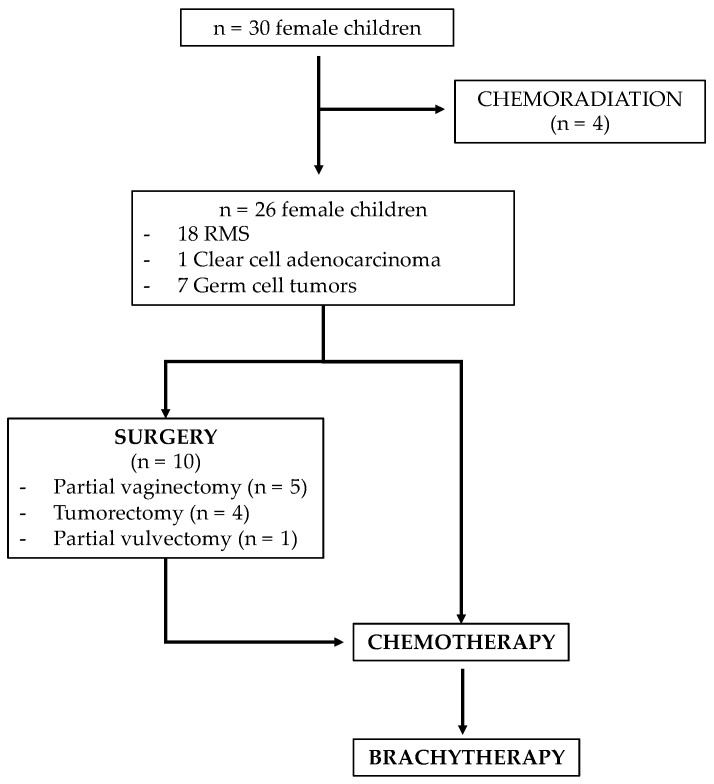
Selection process.

**Table 1 cancers-14-03247-t001:** Patient and tumor baseline characteristics.

Patient and Tumor Characteristics	Variables
Median age (months)	25 (7–147)
Vaginal tumor location	
Upper third	15 (57.7%)
Middle third	3 (11.5%)
Lower third	8 (30.8%)
**Histology**	
Rhabdomyosarcoma	18 (69.2%)
Botryoid	15
Embryonal	3
Clair cell adenocarcinoma	1 (3.8%)
Germ cell tumor	7 (27%)
**Prior chemotherapy**	25 (96.2%)
VA-based	17
Platinum agent-based	7
Doxorubicin-based	1
**Prior vaginal surgery**	10 (38.4%)
Partial colpectomy	5 (50%)
Debulking tumorectomy	4 (40%)
Partial vulvectomy	1 (10%)

Abbreviations: BT: brachytherapy; VA-based regimen were IVA (vincristine, dactinomycin and ifosfamide), VAC (vincristine, dactinomycin and cyclophosphamide) and VAIA (IVA plus doxorubicin).

**Table 2 cancers-14-03247-t002:** Pulse-dose-rate brachytherapy: dosimetric data.

CTV	EQD2, Gy, Median (Range)
Median CTV volume (cm^3^)	2.55 (0.36–11.98)
D 90% (Gy)	51 (9.3–71.4)
D 80% (Gy)	60.9 (0.7–79.3)
**Mean TRAK cGyh^−1^ m^−1^**	1.02 (0.44–2.46)
**Organs at risk**	
Bladder	
D 2 cc (Gy)	24.1 (0.4–51.4)
D 1 cc (Gy)	27.3 (0.4–61.4)
D 0.5 cc (Gy)	35.8 (0.5–85.9)
Rectum	
D 2 cc (Gy)	23.6 (2.4–48.1)
D 1 cc (Gy)	35 (0.3–67.7)
D0.5 cc (Gy)	36.4 (10.4–97.1)
Sigmoid	
D 2 cc (Gy)	3.5 (0.2–24.2)
D 1 cc (Gy)	4 (0.2–28.3)
D 0.5 cc (Gy)	4.45 (0.3–32.2)
**Gynecological structures**	
Ovary (sum of both ovaries)	
D 50% (Gy)	1.2 (0.2–3.2)
D 98% (Gy)	0.7 (0.1–1.9)
Uterus	
D 50% (Gy)	9.8 (0.2–98.1)
Vagina	
D 50% (Gy)	39.7 (0.5–110.8)
D 98% (Gy)	5.6 (0.6–28.3)

Abbreviations: CTV: Clinical Target Volume, TRAK: Total reference air kerma, D 2 cc, D 1 cc, D 0.5 cc: minimal doses delivered to the most 2 cc, 1 cc and 0.5 cc parts of the organ, respectively, D 90%, D 80%: minimal doses delivered to 90 and 80% of the CTV, respectively, EQD2: equivalent doses per 2 Gy fractions.

**Table 3 cancers-14-03247-t003:** Late toxicities according to CTCAE v.5.

Late Effects	Grade		Total = 6
	1–2 *n* = 3	3–4 *n* = 3	
**Gynecologic**			2
vaginal stenosis	0	2	
**Urinary**			2
urinary incontinence	2	0	
**Gastrointestinal**			2
proctitis	1	1	

**Table 4 cancers-14-03247-t004:** Series of low-dose-rate BT for vaginal pediatric carcinomas.

Author	Year	*n*	Vaginal Tumor, *n* (%)	Median Follow-Up (Years)	PFS	OS	Late Toxicities ≥ Grade 2	
							Before 1990	After 1990
Flamant et al.	1990	17	10 (59%)	10	88% (10-y)	88% (10-y)	41.1% (three required surgery)	/
Martelli et al.	1999	38	23 (61%)	5	92% (5-y)	91% (5-y)	NR	NR
Magné et al.	2008	39	26 (67%)	8.4	94.9% (10-y)	82.1% (10-y)	75% (vaginal/urethral sclerosis or stenosis) 15% requiring surgery	20% (vaginal/urethral sclerosis or stenosis) none of them required surgery
Levy et al.	2015	42	41 (98%)	15.5	94% (15-y)	NR	28% G3-4 (all patients)	
Nasioudis et al.	2017	144	74 (51%)	NR	NR	67.1% (5-y)	NR	NR

Abbreviations: PFS: =progression-free survival, OS: overall survival, NR: no reported, y: years.

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
