# Peer review of "Implementation of Image-Guided Brachytherapy for Pediatric Vaginal Cancers: Feasibility and Early Clinical Results"

_cancers, 2022, doi:10.3390/cancers14133247_

Round 1
Reviewer 1 Report
Although it is a rare tumor, the number of patients is too small and too heterogeneous.
In addition, it is unclear whether it is palliative setting or adjunct setting, and information on whether it is a residual tumor is required before BT is performed.
It would be better to compare the treatment performance of 3D RT alone with the group that did not perform RT or the group that performed External RT.
Author Response
Although it is a rare tumor, the number of patients is too small and too heterogeneous.
We thank you for your comment. We agree that this series concerns very few patients but the extreme rarity of this pathology makes it difficult to collect larger data. Genital tumors account for less than 5% of all pediatric neoplasms (than themselves account for 1-2% of all cancers…) Most of these patients are referred to expert centers such as ours and despite the small effective we believe that this series allows us to describe our strategy and the results observed in daily practice. In order to homogenize the data, we have chosen to focus only on vaginal tumors, unlike other series (cited in the discussion) have grouped tumors of the genital tract in the broad sense. For example, in the series by Nasioudis et al. - the largest published to date with 144 cases - 74 patients had tumors of the vagina and vulva without distinction. However, these locations have anatomical and functional particularities that deserve to be considered separately. We also chose to exclude patients who received external beam radiotherapy (EBRT) first, as the prognosis of locally advanced diseases (including N+) is completely different.
In addition, it is unclear whether it is palliative setting or adjunct setting, and information on whether it is a residual tumor is required before BT is performed.
We add this sentence at the end of the section Introduction: “In this series, we examined the outcome of girls treated with image-guided pulsed dose rate BT (PDR-BT) for a vaginal tumor with curative intent.” The curative intent was also specified in the section Material and Methods. Moreover the section Material and Methods specified that brachytherapy was performed “as part of an organ-preservation strategy”.
The manuscript detailed in the section 2.2. Treatment the choice of local treatment according to the histology. In general, BT was chosen when surgery was difficult to perform or if associated with a risk of morbidity.
It would be better to compare the treatment performance of 3D RT alone with the group that did not perform RT or the group that performed External RT.
We thank you for this suggestion. The management of these patients is classically based on a multimodal strategy; there is therefore no indication for EBRT alone in a curative intent and to our knowledge no series in the literature reports the results of patients receiving EBRT without BT. Similarly, patients receiving EBRT first have more advanced diseases with a worse prognosis (e.g. clear cell histologies rather than rhabdomyosarcoma). In the inclusion period of our study (from 2005 to 2020) only four patients received EBRT before brachytherapy. This very low number of cases would not have allowed a comparison with patients without prior EBRT (n=26). Finally, while our series is interested in the rate of long-term toxicities, the consideration of patients with EBRT would have complicated the interpretation of the results.
The following sentence was added at the end of the section Discussion: “Finally, we chose to exclude patients who received chemoradiationbecause of locally advanced diseases associated with a worse prognosis and because EBRT treatment is associated with different toxicities that would have biased the interpretation of the results.”

Reviewer 2 Report
The manuscript proposed by Terlizzi et al. It deals with an extremely delicate issue, such as that of an innovative method of dispensing the dose of brachytherapy in a female pediatric population suffering from rare oncological diseases of the lower genital tract, which put young patients at risk, in case of survival, of infertility. of sexuality disturbance and altered body image. Obviously, the rarity of the disease raises the problem of only an indirect and historical comparison with other brachytherapy methods previously used by the same group of researchers.
The method is well described, the results are presented both as therapeutic success and as a reduction in complications.
As the only observation I would suggest checking the spelling of some terms especially in table 1.
Author Response
The manuscript proposed by Terlizzi et al. deals with an extremely delicate issue, such as that of an innovative method of dispensing the dose of brachytherapy in a female pediatric population suffering from rare oncological diseases of the lower genital tract, which put young patients at risk, in case of survival, of infertility. of sexuality disturbance and altered body image. Obviously, the rarity of the disease raises the problem of only an indirect and historical comparison with other brachytherapy methods previously used by the same group of researchers.
The method is well described, the results are presented both as therapeutic success and as a reduction in complications.
As the only observation I would suggest checking the spelling of some terms especially in table 1.
Thank you for your comment. We have clarified some of the terms in Table 1. The word “mild” has been changed to “Middle” and “vaginectomy” to “colpectomy”.
